# Community Seedbanks in Uganda: Fostering Access to Genetic Diversity and Its Conservation

Rose Nankya [1,*], Abdel Kader Naino Jika [2], Paola De Santis [2], Hannington Lwandasa [3], Devra Ivy Jarvis [2,4,5] and John Wasswa Mulumba [3]

[1] Alliance of Bioversity International and CIAT, Uganda Office, Kampala P.O. Box 24384, Uganda
[2] Alliance of Bioversity International and CIAT, 00153 Rome, Italy; a.naino@cgiar.org (A.K.N.J.); p.desantis@cgiar.org (P.D.S.); d.jarvis@cgiar.org or d.jarvis@agrobiodiversitypar.org or dijarvis@wsu.org (D.I.J.)
[3] National Agriculture Research Organization of Uganda, Entebbe P.O. Box 295, Uganda; lwahann@gmail.com (H.L.); jwmulumba@yahoo.com (J.W.M.)
[4] Platform for Agrobiodiversity, Museo Orto Botanico di Roma-Universita' La Sapienza, Largo Cristina di Svezia 23A-24, 00165 Rome, Italy
[5] Department of Crop and Soil Sciences, Adjunct Faculty, Washington State University, Pullman, WA 99163, USA
[*] Correspondence: r.nankya@cgiar.org; Tel.: +256-782-574-916

**Abstract:** Community seedbanks promote conservation and the use of crop genetic diversity, as well as supporting farmer seed systems. This study analyses seed flow and access to crop genetic diversity over time in the Nakaseke, Rubaya, and Kibuga seedbanks of Uganda. The modes of operation of the banks were compared through scrutinizing records of crops and varieties being conserved, quantities of seed distributed, to whom, and quantities returned. The Nakaseke seed bank distributed the highest varietal diversity of common bean (*Phaseolus vulgaris*) and groundnuts (*Arachis hypogaea* L.), whereas the Rubaya seedbank distributed the highest quantity of common bean seed, followed by the Kibuga seedbank. There were no significant differences between the type of variety of seed, quantities of seed accessed, and seed returned to the seedbanks by women and men—except for the Nakaseke seedbank, where women returned significantly higher quantities of common bean seed. The Kibuga and Rubaya seedbanks dealt with individual farmers, whereas the Nakaseke seedbank dealt with individual farmers and groups. The extent to which core functions were achieved by a particular seedbank depended on the mode of operation, including actors, management, degree of development, socio-economic setting, among others. Further research is recommended to unpack these factors and come up with the most appropriate combinations for greater seedbank effectiveness.

**Keywords:** community; seedbank; crop; genetic; diversity; conservation; use; core; roles

## 1. Introduction

The Convention on Biological Diversity (CBD) Global Biodiversity Framework sets out an ambitious plan to transform society's relationship with biodiversity by 2050, placing biodiversity at the core of the work being done to promote sustainable food systems. Feeding and truly nourishing humanity in the face of climate change depends on the world's small-scale farmers maintaining and developing agricultural biodiversity. As well as being essential for the resilience and stability of agricultural production systems, agricultural biodiversity is fundamental to the livelihoods, health, and nutrition of billions [1]. The Green Revolution in agriculture, and the globalisation of food systems, have led to the rapid decline of agrobiodiversity [2]. In recent decades, the focus on formal breeding coupled with the homogenization and reduction in number of seed companies, and the functioning of global markets, have led to the promotion of the uniform planting of single varieties at the expense of access to crop variety diversity [3]. This trend has ignored fundamental

issues concerning the availability and use of crop diversity by smallholder farmers, who often have distinct needs for a wider range of diversity that is adapted to their vulnerable ecosystems [4]. Many smallholder farmers in developing countries have limited access to adequate and diverse planting materials that are needed to improve the production of their staple crops [5]. Related to the importance of farmers' seed, is the maintenance of varieties by farmers, including both improved and local varieties [6].

Farmers' seed systems have a wider significance than just the supply of seed and maintenance of local varieties (landraces); they constitute a dynamic in situ conservation system which allows the continued evolution of crops. Farmers' seed systems have a key role in the global management of plant genetic resources for food and agriculture, as pointed out by the Food and Agriculture Organization (FAO) of the United Nations [7]. According to Borges et al. [8], improved varieties were found to complement local varieties rather than completely replacing them, resulting in increased on-farm biodiversity. When asked by researchers why they retained their local varieties, farmers often said, "Our varieties are well adapted to our local conditions," underlining the resilience value of the landrace material [9]. The retention of landraces is common in marginal agriculture for risk avoidance through the maintenance of heterogeneity and for the conservation of desirable varietal attributes (regarding cultivation, consumption, and cultural aspects) [10,11].

Community seedbanks (CSB), also called community genebanks, have come to the fore lately, as they are key in promoting crop genetic diversity conservation and use. These "banks" are collections of seeds that are maintained and administered by the communities themselves. Seeds can be stored by a community either in a large quantity to ensure that planting material is available for distribution, or in small samples to ensure that genetic material is conserved and available for multiplication should varieties become endangered [12]. For instance, the South African government established and supported a national network of community seedbanks in the country to strengthen farmers' seed systems, support conservation of farmers' varieties, maintain seed security at district and community levels, and invite farmers to participate in making national-level decisions related to the conservation and sustainable use of plant genetic resources [13,14]. In the same way, the Green Foundation motivated members of the local Krushi self-help groups to establish community seed banks in a selected cluster of villages in India [12]. Many other community seedbanks are documented in different countries around the world [15]. Operation models of seedbanks are flexible and vary from seedbank to seedbank. In general, at the start of the planting season, farmers can borrow seeds of specific varieties which will have to be returned after harvesting. The return rate is established in the constitution, and it represents a form of profit (still in terms of seeds) for the seedbank and guarantees a seed stock. For Kibuga and Nakaseke seedbanks, the return rate is 50% of the quantity of seeds borrowed, whereas for Rubaya, it is 30%. The quantity of seeds returned to the bank after every growing season, reflects the extent to which the farmers abide by the seedbank constitution, and the extent of the growth of the seedbank in terms of the quantity of seed contained in that bank. If a farmer fails to return the seeds due to a genuine reason (such as drought), the return can be postponed to the following season. In case of failure, the famer can be requested to pay cash, or can be reported to local authorities and forced to pay. Some studies have shown that a robust local community that manages locally adapted seeds, a diversity of crops, and has strong local institutions such as national research organizations and agriculture extension services, has a better chance of adapting production systems to changing conditions such as climate change [12].

As noted by Sthapit and colleagues in Nepal [16], the community seed/gene banks differ in terms of actors and objectives, origin and evolution, functions and activities, governance, management, cost, technical operations, support and networking, policy and legal environment, as well as sustainability. The number, diversity, and degree of development in terms of community seedbanks vary greatly among countries; some countries have a total ranging from 40 to more than 100, whereas other countries have a formal national network of community seedbanks or local seed-saver groups [17]. In

Uganda, the community seedbanks are at different stages, and are mostly at the early stages of their development. Against this background, the study aimed at assessing how three community seedbanks, in three sites of Uganda, namely, Nakaseke, Rubaya and Kibuga, enabled: (i) farmers' access to crop genetic diversity including local, creole, and improved varieties; (ii) access to crop genetic diversity in terms of two gender groups, male and female; and (iii) seed sovereignty in their respective communities. The modes of operation as key drivers of success and failure in those core functions of community seedbanks [17] were discerned to compare their effectiveness. This study moved community seed banking from the conceptual and theoretical sharing of experiences to assessing and analyzing the practicalities and realities of the concepts and theories in the three banks. The findings contribute to the emerging knowledge base and experiences, which is crucial for the much required technical, financial, and political support, and is necessary for the proliferation of more effective community seedbanks, since their services are needed now more than ever.

## 2. Materials and Methods

In this study, we analyzed records of crops and varieties managed by the community/seed/genebanks in terms of their seed flow (quantities of seed distributed, returned seed, and stocks conserved). Access was assessed in terms of the type of variety of seed, and quantity of seed that farmers borrowed from seedbanks in each of the seasons during the study period. This access reflected how farmers benefited from the seedbanks. Access was discerned by gender to understand whether women benefited from the seedbanks to the same extent as men, and whether a preference for certain varieties was influenced by gender. Seed sovereignty was gauged from the way that the farmers' right to breed and exchange of diverse seed was enabled by the seedbanks, which forms the backbone for crop genetic diversity conservation. The modes of operation of seedbanks were analyzed regarding how they influenced the varieties and quantities of seeds distributed/accessed from the seedbanks, as well as the quantities and varieties returned, in order to understand how they affected the extent to which the core functions of seedbanks were achieved. Furthermore, the extent to which seeds were being returned to the seedbanks was also analyzed. The members of the seedbanks have a good knowledge regarding the common bean and groundnut varieties managed in the seedbanks and can identify them. This is because most of the farmers have been involved in biodiversity projects that worked on the characterization of these varieties through focus group discussions, on-farm characterization and selection, as well as through diversity fairs. These projects have been implemented by Bioversity International, in partnership with the National Agriculture Research Organization of Uganda, since 2008. The varieties that this study focused on are edible for human beings and animals. They are listed in the supplementary materials for each seedbank, Data for the manuscript entitled Community Seedbanks Core Roles of Fostering Crop Genetic Diversity Conservation and Use Achievable; the Case of Seedbanks in Uganda. Zenodo. Available online: https://zenodo.org/badge/DOI/10.5281/zenodo.6013093.svg (accessed on 8 February 2022).

The Nakaseke seedbank is in the Nakaseke district of the central region within the Central Wooded Savannah agro-ecological zone. The area is in an altitudinal range of 1086–1280 m a.s.l., with mean annual rainfall of up to 1100 mm and temperatures ranging from 16 °C to 30 °C. The Rubaya and Kibuga seedbanks are in the Kabale district in the south-western highlands. This area has an altitude ranging from 1800 m to over 2200 m a.s.l., with a mean annual rainfall of up to 1100 mm, and temperatures ranging from 11 °C to 25 °C [18] (Figure S1). The Nakaseke and Kibuga community seedbanks have been recently established, in 2014 and 2018, respectively, they are operated communally by farmers that started working together with the construction of the seedbanks, and their groups are in the formative stage. As a result of the fact that the groups are just starting to evolve, the membership keeps fluctuating. The membership seems to be linked to the harvest; after good seasons, farmers do not go back to the bank to borrow seeds. On the other hand, the Rubaya community seedbank was established in 2012, but the Kagyera

Bataka Farmers' group operating it has been in existence since 1970, and its membership does not fluctuate (Table S1).

The data was entered into Excel and analyzed using R software developed by the Core Team of 2013 [19]. We first carried out descriptive statistics (mean, minimum, maximum, standard error, first and third quartiles) for each seedbank and each species within the seedbank. The data was tested for normality using the Shapiro–Wilk test, and for homogeneity of variances using the Levene test; however, even after transformation, the data violated the conditions for application of the parametric ANOVA. Kruskal–Wallis nonparametric ANOVA [20], and its extension for two way ANOVA (Scheirer–Ray–Hare test) [21,22], were then used to determine the significance of the differences observed among and within the seedbanks. Multiple comparison was done using the Pairwise Wilcoxon test and adjusted as suggested by Holm [23].

## 3. Results

### 3.1. Crop GENETIC Diversity Handled by the Community Seedbanks

3.1.1. Quantities of Seeds Distributed

The three seedbanks conserved and distributed seeds of different varieties of common bean (*Phaseolus vulgaris*), although, only the Nakaseke seedbank conserved and distributed seeds of different varieties of groundnuts (*Arachis hypogaea* L.). The Rubaya seedbank distributed the highest quantity of common bean seeds, and therefore, had the highest mean. Despite being operational for only two years (2019–2020), the Kibuga seedbank was second in terms of the quantity and mean quantity of seeds distributed. Nakaseke distributed the highest number of seed lots (quantity of seed given out at a particular point in time), implying that it served the highest number of beneficiaries compared with the others. In the same way, the Rubaya seedbank had the highest mean for the quantity returned, followed by the Kibuga seedbank, and the lowest was for the Nakaseke seedbank, whereas the reverse is true for the mean balances (Table S2, Figure S2).

There are two growing seasons in Uganda for annual crops, including the common bean and groundnuts. The seedbanks differed in the quantities of common bean seed distributed every season (Kibuga vs. Nakaseke: $p = 0.000$; Kibuga vs. Rubaya: $p = 0.000$; Rubaya vs. Nakaseke $p = 0.000$). In the Rubaya seedbank, the second season of 2017 had the highest quantity distributed; however, this difference is only significant compared with the 2020 first season. The quantity of groundnut seeds distributed every season also differed significantly ($p = 0.00$), except in the two seasons of 2020. The quantity for the second season of 2019 was highest (Figures S3 and S4).

3.1.2. Types of Varieties Distributed by the Seedbanks

In this study, varieties were classified into local/landrace, modern, and creole. A landrace is a domesticated, locally adapted, traditional variety of a species of a plant that has developed over time, and has adapted to its natural and cultural environment. It is a cultivar that has been improved by traditional agricultural methods [24,25]. Modern varieties, also considered as improved, are purposefully selected plant strains— "scientifically bred to be uniform and stable" [26]. Creole varieties are formally released varieties that have been planted in the farmer's fields for many years in local environmental conditions and selection processes, and have therefore lost their original genetic makeup. "Creole landrace may be derived from an originally bred variety, which then becomes an effective landrace following numerous repeated cycles of planting and farmer seed selection in a specific location" [27,28].

The seedbanks differed in terms of the type of variety of common bean and groundnut seed distributed. The largest quantity of groundnut seeds distributed by the Nakaseke seedbank was for the local varieties, rather than the modern varieties. Regarding the common bean, the Rubaya and Kibuga seedbanks distributed larger quantities of modern varieties, whereas Nakaseke distributed the largest quantity of creole varieties, followed

by local varieties. Rubaya distributed the second largest quantity of creole varieties, and Kibuga did not distribute any creole varieties (Figure S5).

### 3.1.3. Quantities of Seed Returned to the Seedbanks

The weather conditions that prevail in a particular season affect the yield positively or negatively, and this has implications for the quantities and varieties that get returned to the seedbanks after harvesting. For instance, the amount of rainfall, and the way it is distributed throughout the two growing seasons of each year, cannot be the same in this era of climate change (Table S3).

The quantities of common bean seeds returned to the banks by farmers differed significantly ($p = 0.000$) among seasons for the Nakaseke and Rubaya seedbanks. The quantity of groundnut seeds returned to the Nakaseke community seedbank differed significantly ($p = 0.000$) among the years and seasons, except for the second season of 2019 which did not differ significantly from the two seasons of 2018, and the second season of 2020. The quantity returned in the first season of 2018 was, on average, the highest, as the other seasons were bad (Table S4).

For the common bean in the Nakaseke seedbank, the quantity of seed returned in the second season of 2017 was, on average, higher than the other seasons; however, the difference is only significant compared with the two seasons of the year 2020 (2017 S2 vs. 2020 S1: $p = 0.007$; 2017 S2 vs. 2020 S2: $p = 0.02$). In the Rubaya seedbank, the quantity of common bean seeds returned in the two seasons of 2020 differed significantly ($p = 0.000$) from the other seasons, except the second season of 2019, for which the difference is not significant (Figure S6).

### 3.2. Access to Crop Genetic Diversity by the Two Gender Groups: Males and Females

The percentage of female to male farmers that were involved in community seed banking, and were therefore focused upon in our study, varied between the seedbanks. In Rubaya, the percentage of males was equal to that of females, whereas in Kibuga, the percentage of males was half that of females; however, for Nakaseke, the percentage of males was almost one third of the females. This is reflected in Table S5 of the results, which shows the seedlots as taken by the two gender groups and in the Supplementary Materials, Data for the manuscript entitled Community Seedbanks Core Roles of Fostering Crop Genetic Diversity Conservation and Use Achievable; the Case of Seedbanks in Uganda. Zenodo. Available online: https://zenodo.org/badge/DOI/10.5281/zenodo.6013093.svg (accessed on 8 February 2022).

The Rubaya seedbank had more equitable access to the seeds by both gender groups, as compared with the other seedbanks (Table S5; Figure S7).

Gender did not significantly affect the types of common bean varieties accessed, although, preferences for certain types were realized. In the Nakaseke seedbank, female farmers preferred local varieties, even though the quantities taken by women were not significantly higher than those taken by men. Men preferred creole varieties, although, the quantities taken by men were not significantly higher than those taken by women. For modern varieties, there were no differences in the preference and quantities taken by women and men. In Rubaya, both men and women preferred modern varieties, followed by local varieties for men; however, creole varieties followed for women. Regarding the amount of seeds returned, there was a significant ($p = 0.006$) effect in terms of gender for the common bean, whereby women returned significantly higher quantities compared with men. On average, male farmers returned significantly higher quantities of local common bean varieties compared with females (4.7 kg vs. 2.7 kg, respectively; $p = 0.02$). For modern varieties, females returned a significantly ($p = 0.002$) higher average quantity than males (8.4 kg for females and 7.8 kg for males). For groundnut seeds, gender did not significantly influence the quantity and type of seeds accessed or returned, although, the female farmers returned greater quantities of seed compared with men. No gender effect was realized in the Kibuga seedbank at all.

*3.3. Modes of Operation of the Seedbanks and Their Effectiveness in Recovering the Loaned Seeds*

The three seedbanks were different in their modes of operation in that the Kibuga and Rubaya seedbanks loaned seeds to individual farmers, whereas the Nakaseke seedbank loaned seeds to both individual farmers and groups. The quantity of seeds for the common bean returned by individual farmers differed significantly ($p = 0.03$) from that returned by groups, in that individual farmers returned a significantly greater quantity of seeds than the groups. Significant differences were also revealed for the returning rates of individual farmers vs. groups by variety type ($p = 0.000$) whereby the individual farmers returned a significantly higher quantity of modern and creole varieties, whereas the groups returned a significantly greater quantity of local varieties. Other quantities that were significantly higher than their counterparts were: the quantity of local and creole varieties for the groups ($p = 0.01$); quantity of creole varieties for individual farmers; the quantity of local varieties for groups ($p = 0.000$); and the quantity of creole and local varieties for individual farmers ($p = 0.000$), respectively. Regarding the groundnut seeds, the quantity of seed taken and returned by the groups was significantly ($p = 0.000$) greater than the quantity taken and returned to the seedbank by individual farmers. The differences in the quantities of seed that were defaulted were not significant among the seedbanks.

## 4. Discussion

*4.1. Crop Genetic Diversity Handled by the Community Seedbanks*

Community seedbanks increase farmers' access to crop and seed diversity, as was seen in the Welthungerhilfe program community seedbanks in Kirundo [29]. They are designed as a strategy for ensuring local seed security and germplasm conservation [30]. The three seedbanks conserved and distributed seeds for different varieties of common bean (*Phaseolus vulgaris*), whereas only the Nakaseke seedbank conserved and distributed seeds of different varieties of groundnuts (*Arachis hypogaea* L.). This is in accordance with the three core functions of seedbanks identified in [18], namely: conserving genetic resources; enhancing access to and availability of diverse local crops; and ensuring seed and food sovereignty. The studied community seedbanks served communities to different extents, as shown by the quantities of seed they distributed, and the number and type of variety of seed, which still falls in their documented mandate of maintaining seeds for local use [12,15,16]. Rubaya seedbank distributed the largest quantity of common bean seeds followed by Kibuga seedbank and Nakaseke seedbank ( supplementary material). However, Nakaseke had two times the number of common bean varieties (thirty one varieties) compared to Rubaya seedbank (fourteen varieties) and four times the number of varieties conserved in Kibuga seedbank (seven varieties). In the same way, Nakaseke seedbank distributed the greatest number of seed lots having had the highest number of beneficiaries. The number of varieties in a seedbank seemed to be inversely proportional to the quantity of seeds produced and distributed by the same bank.

The capacity of seedbanks in managing and enabling access to diversity depends on several factors. In this study access to diversity is reflected by the quantity of seed of particular species and varieties distributed to farmers by the community seedbanks. Access to land [31] as well as availability of labor, machinery, and financial resources, can influence the number of varieties multiplied and managed, including seed quantity per type of variety distributed by the seedbank. The arable land available in the two districts of the study (Kabale for Rubaya and Kibuga seedbanks; Nakaseke for Nakaseke seedbank) differs significantly; the population density in Kabale equals 361 people per sq. km [32] against the 67.5 people per sq. km [33] in Nakaseke. Access and availability of land plays an important role in selecting the type of variety produced: farmers in Rubaya and Kibuga seedbanks mainly grow climbing common bean varieties that are suited to areas with limited arable land [34], whereas farmers in Nakaseke grow common bean bushy varieties (Table S1). This choice influences the availability of seed per variety, as climbing bean varieties can produce up to 4–5 tons per ha, compared with 1–2 tons ha −1 of bush bean

varieties in Uganda [35]. This could explain why the seed quantities in the Rubaya and Kibuga seedbanks are higher than those of the Nakaseke seedbank.

That aside, the Rubaya seedbank is operated by a farmers group that has been in existence since the 1970s; however, the other two seedbanks are operated by farmer groups that came into existence with the construction of the seedbanks (2014 for Nakaseke, and 2018 for the Kibuga seedbank), which shows the extent of development for these seedbanks. The unstable membership of the Kibuga and Nakaseke seedbanks does negatively affect the quantities handled by the seedbanks, thereby confirming that the extent of development of the community seedbank affects the effectiveness of the seedbank.

The community seedbanks in this study tended to focus more upon improved and creole varieties of common bean, and this is contrary to what is noted in [17], which reports that generally, community seedbanks store local/farmer varieties. Rubaya and Kibuga seedbanks distributed more seed of modern varieties, whereas Nakaseke distributed the most seed for creole varieties, followed by local varieties. This could be due to the lack of market for farmer varieties (local varieties), as is the case with many other countries that uphold restrictive laws, such as seed certification based on the criteria of the formal seed system of distinctness, uniformity, and stability [12], to which some local varieties do not conform. It may also be caused by government programs that prioritize and distribute modern/improved varieties in the formal seed system, excluding the farmers' local varieties, as well as training and orientating extension agents that favor the promotion of improved varieties at the expense of farmers' varieties. The factors above may make farmers' local varieties/landraces look inferior to modern/improved varieties; however, they are not inferior at all in many attributes [12].

The quantities of seeds returned to the banks differed significantly ($p = 0.000$) among seasons, which is most likely due to the weather patterns that prevailed during the different seasons. The changing weather patterns affect the growth and yield of different varieties of crops, and bad seasons usually result in lower returning rates, as documented for the Kiziba seedbank [36].

### 4.2. Access to Crop Genetic Diversity and Seed Returning Rates by the Two Gender Groups: Male and Female Farmers

Table S5 shows that women were more active than men in borrowing seeds, as they took more seedlots than men in all the seedbank; however, gender did not significantly affect the types and quantities of common bean and groundnut varieties, and the seeds accessed implied that the varieties and quantities of seeds accessed were not determined by gender. However, gender significantly affected ($p = 0.006$) the quantity of common bean seeds returned in Nakaseke, whereas on average, women returned a higher quantity compared with men. For groundnut seeds, gender did not significantly influence the quantity and type of seeds returned, although, the female farmers still returned greater quantities of seed compared with men; however, some gender imbalance was realized in returning seeds, but generally, community seed banking enabled men and women to work together. Earlier studies noted that women are always more active than men [37,38], and that they meet necessary obligations more than men. In this study, this is shown by the significantly higher amounts of seed returned by the women compared with men. The commitment of women in borrowing and returning seeds to the banks is also in accordance with what was observed in another study, whereby the number of women in farmer groups managing community seed banks outweighed that of men and their knowledge on seed storage, aptitude for nurturing with patience, and ability to save seeds for future seasons, often made women better than men in managing seed banks [12]. Sadly, women's roles are not always recognized by formal-sector agencies [17]. The best option would be for community seedbanks to formally recognize women, and to appreciate and support their active role, knowledge, passion, and expertise in managing seedbanks.

*4.3. The Modes of Operation of the Seedbanks and Their Effectiveness in Recovering the Loaned Seeds*

The three seedbanks were different in their modes of operation in that the Kibuga and Rubaya seedbanks loaned seeds to individual farmers only, whereas the Nakaseke seedbank loaned seeds to both individual farmers and groups. This agrees with what was documented earlier—that community seedbanks vary in management models and administration [39]. The quantity of the common bean seed returned by individual farmers was significantly higher (*p* = 0.02) than that returned by groups. The individual farmers returned a significantly higher quantity of modern and creole varieties of common bean, whereas the groups returned a significantly greater quantity of local varieties than the individual farmers. This could imply that the group approach is more effective in conserving local varieties than the individual farmer approach. The quantity of groundnut seeds taken and returned by groups was significantly greater than the quantity taken and returned to the seedbank by individual farmers; however, the differences in the quantities of seed that were defaulted by individual farmers and groups were not significant among the seedbanks, which could imply that the two approaches are not significantly different in meeting their seed banking obligations.

**5. Conclusions**

This study has moved community seed banking from the conceptual and theoretical sharing of experiences, to showcasing the practicalities and realities of the concepts and theories. The results highlight the contribution of community seedbanks in securing farmers' access to crop and seed diversity of local varieties/landraces, and improved/modern and creole varieties. There was heterogeneity in the focus of the studied seedbanks on the different types of varieties, whereby the larger quantity of groundnut seed distributed by the Nakaseke seedbank was for local varieties; however, for the common bean, the Rubaya and Kibuga seedbanks distributed larger quantities of modern varieties, and Nakaseke distributed the largest quantity of creole varieties. This heterogeneity is good in that it enables the seedbanks to serve farmers under different socio-economic conditions. Since the content of the seedbanks influence what is requested and accessed by farmers, seedbanks should be managed in such a way that they meet farmers' demand in terms of quantities and varieties; however, seedbanks should put greater emphasis on conserving landraces to prevent their extinction, as they are important sources of genetic material for breeding due to their wider adaptation to different conditions that enables them to withstand both abiotic and biotic stresses [35]. Community seed banks need to establish demonstration gardens and/or on-farm trials in their areas of operation in order to assess the yield of landraces, creole, and improved varieties on the same plot under the same management conditions to enable farmers make the best variety choices for their conditions. With regard to widening the variety choice base, seedbanks should consider expanding their variety collections through networking with public national genebanks, local and international seedbanks, but also through other social networks. There is a need to improve the capacity of seedbanks in research and networking to enable more effective management of the crop genetic resources they handle.

The study has revealed that women are more enthusiastic and active in community seed banking, and they should be formally recognized, appreciated, and supported for their active role, knowledge, passion, and expertise in managing seedbanks. The results also show that the extent to which the core functions of genetic resource conservation and seed sovereignty were achieved, depended on the modes of operation of the seedbanks, including actor management, degree of development, and socio-economic setting, as documented in [40]. Further research is recommended to unpack each of these factors, and to come up with the appropriate combinations of these factors that make the most conducive environment for effective community seed banking in particular contexts. In order to have more comprehensive information on the role of community seed banks, further research could include a social study by interviewing farmers to understand how

they value seedbanks, and how they would like to improve them to complement what is captured in the approach of our study.

**Supplementary Materials:** The following supporting information can be downloaded at: https://www.mdpi.com/article/10.3390/resources11060058/s1, Figure S1: Map of Uganda showing where Nakaseke, Kibuga and Rubaya community seedbanks are located; Figure S2: Box plots of the mean quantity of common bean seed (kg) distributed by each seed bank over the study period (Nakaseke and Rubaya 2017–2020; Kibuga 2018, 2019 and 2020); Figure S3: Box plots of the mean quantity of common bean seed (kg) distributed by the seedbanks in each of the two growing seasons from the year 2017 to 2020; Figure S4: Box plots of the mean quantity of groundnut seeds (kg) distributed by Nakaseke seedbank in the two growing seasons of 2018, 2019 and 2020; Figure S5: Box plots of the mean quantity of common bean seed (kg) of each variety type distributed by the seedbanks; Figure S6: Box plots of the mean quantity of common bean seeds (kg) returned every season to the respective community seedbanks during the study years; Figure S7: Box plots of the mean quantity of seeds (kg) distributed by gender in each seedbank; Table S1: Characteristics of the studied Community Seed Banks; Table S2: Summary statistics of the quantities of seeds distributed, returned and balances of seed that had not yet been returned to the seedbanks over the study period; Table S3: The monthly rainfall totals in mm of rainfall received in Nakaseke from January 2019 to August 2021; Table S4: Quantity of groundnut seeds (kg) borrowed and returned in the respective seasons in Nakaseke seedbank; Table S5: Summary statistics about the access to seed by the two gender groups.

**Author Contributions:** Conceptualization, R.N., P.D.S., H.L. and J.W.M.; data curation, A.K.N.J.; formal analysis, A.K.N.J.; funding acquisition, D.I.J.; investigation, R.N. and H.L.; methodology, R.N., H.L. and P.D.S.; project administration, R.N. and H.L.; resources, D.I.J. and J.W.M.; software, A.K.N.J.; supervision, D.I.J. and J.W.M.; validation, P.D.S.; writing—original draft, R.N.; writing—review and editing, R.N., A.K.N.J., P.D.S., D.I.J. and J.W.M. All authors have read and agreed to the published version of the manuscript.

**Funding:** This research was funded by the International Fund for Agricultural Development (IFAD) Grant number 2000001629, "Use of genetic diversity and evolutionary plant breeding for enhanced farmer resilience", together with the in-kind contribution from the Government of Uganda in terms of staff time of the National Agriculture Organization of Uganda staff, office space and field assistance. The APC was funded by IFAD.

**Institutional Review Board Statement:** Not applicable.

**Informed Consent Statement:** Not applicable.

**Data Availability Statement:** The data presented in this study are openly available in the article and supplementary materials and on request from the corresponding author.

**Acknowledgments:** We thank the site teams and the farmers of the Nakaseke, Rubaya and Kibuga seedbanks for the provision of the community seedbank records upon which the paper is based.

**Conflicts of Interest:** The authors declare no conflict of interest. The sponsors had no role in the design, execution, interpretation, or writing of the study.

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
