# Peer review of "Community Seedbanks in Uganda: Fostering Access to Genetic Diversity and Its Conservation"

_resources, doi:10.3390/resources11060058_

Round 1

Reviewer 1 Report

General:

How are these different bean varieties identified? It should be included in the methods section, reporting if all are edible or have also other use? (for instance as feed for animal, or other?). It would be helpful for the clarity of the article to also list the varieties names per studied area in a short table (or in supplement as well).

Also, the process should be explained the farmers receive the seeds and then return them. For instance, in the manuscript, it is stated that seeds were borrowed and then returned, i.e. if farmers borrow and then are obliged to return the same or double amount they borrowed (is this actually demonstrated in table S1?) after harvesting. What happens when the farmers are not able to return this? Do they pay?

Why did the authors choose to include gender in their study? This should be also clearly justified (what is the % of male/female population that are farmers?). Please see specific comments below on this topic (e.g. relation of gender with accessibility, women empowerment, seeds return etc.)

Lines 44-45: please add references.

Lines 87-89: please add references.

Line 99: please indicated (for instance in parenthesis) what kind of local institutions (e.g. research centers, or other?).

Line 154: please add reference for R.

Line 168, 305: L should not be in italics (Arachis hypogaea L)

Line 169: not so clear, “the quantity taken…” what is meant? Please rephrase more clearly.

Lines 183-184: rather the boxplots are for the period between 2017-2020, as we do not have a different boxplot for each year as implied in the label of the figureS2.

Line 325: how was access considered in the current study, as is known from literature that access can be a factor of influence? It should be explained in the methods section why the specific variables were chosen to be investigated.

Line 351: “ look inferior…” not comprehensible, what is meant? Please recheck and explain clearly.

Lines 372-375: please justify with references. Also, where in the manuscript is it shown that the number of female farmers is higher than the male ones, in each region? Please clearly demonstrate data that support statements.

Lines 405-408: it is questionable how this was demonstrated in the current study. Where such negative bias was set as an issue in the manuscript, and more specifically in introduction? Please recheck and clarify.

Line 411: please provide e.g., via reference, what is the state of women empowerment in this sector and in the studied areas?

Author Response

Dear Reviewer, please find the attached file bellow. 

Reviewer 2 Report

I highly appreciate the topic of the paper and think that this article has strong potential. However, it needs a moderate amount of work before being published.

Introduction. The authors introduce the main concept of community seed banks relatively late (already on page 2). I think in the second paragraph of the introduction you could already introduce community seed banks and state what the goal of the article is.

Conclusion. The Conclusion does not quite match the results of the study and is not well developed. As I understand it, the purpose of the study is to better comprehend the functioning of the seedbanks, which provide both traditional and improved varieties (as well as creole varieties), and to provide some recommendations.

Another note: while earlier the authors discuss how the addition of improved varieties (alongside traditional varieties) can help to increase crop diversity and resilience, in the conclusion they instead focus almost entirely on traditional varieties - in spite of the farmers' revealed demand for improved varieties in the seedbank distribution data.

Last, on a broader level, while the statistics and numbers provided are useful, I feel that they could be greatly improved by interviews of the farmers, to ask them how they use the seedbank and see its value, how they see its strong points and also how they could see it improved. It is hard to infer too much from the numerical data and I feel that including at least some interviews of farmers would greatly enhance the insights provided by the study, though perhaps this is beyond the scope of this article.

Author Response

Dear reviewer, please find the attached file with replies. 

Reviewer 3 Report

Some deductions and conclusion are a little straightforward or insufficiently explained/justified. For the rest, the paper is very interesting and timely. See my comments in the document for more detailed suggestions.

Author Response

(The authors gave the same response as above.)

Round 2

Reviewer 2 Report

The revision is good. Thanks for adding the suggestion for a social study featuring interviews as a next step in the conclusion.

For the conclusion, I am still not happy with how it is stated (I don't think it matches your findings). "The seed banks in this study tended to focus more on improved and creole varieties;" but for example for Nakaseke 22 out of 21 bean varieties conserved were local. So I would highly recommend a bit more nuance here. "The largest quantity of groundnut seeds distributed by Nakaseke Seedbank was for local varieties (line 245)." Please highlight the heterogeneity here between the three gene banks.

You could also mention that the contents of the seed banks in a way influence what is requested. In the conclusion you should outline your philosophy for how the seed banks should work. That is, should seed banks focus on 1) matching the demand of farmers? or 2) focus on conservation of local varieties or 3) some combination of these two? How should the seed banks consider expanding their collections? And provide more of a choice to farmers by for example running trials of creole, improved and landrace varieties of a given crop in the same plot to demonstrate their performance for farmers in the village?

Also, here are a few text edits: "Green Revolution" should be capitalized on line 43; "cultural" on line 68; "Access to" on line 112; "seed banks" on line 458. Please re-read carefully to proofread.

Author Response

20th May 2022

The Editor,

Resources Journal

Dear Sir/ Madam,

RE:  Second Round Revision of the Manuscript resources-1610452, ‘Community Seedbanks Core Roles of Fostering Crop Genetic Diversity Conservation and Use Achievable; the Case of Seedbanks in Uganda’

The authors of the above manuscript are very grateful for the second round of very constructive comments given by the reviewer 2 that have enabled us to improve further the quality of the manuscript. We hereby respond to the comments raised by the reviewer and explain point by point, the details of the revisions made in the manuscript in accordance to the responses; as below:

Reviewer 2 – Review Report Round 2

For the conclusion, I am still not happy with how it is stated (I don't think it matches your findings). "The seed banks in this study tended to focus more on improved and creole varieties;" but for example for Nakaseke 22 out of 21 bean varieties conserved were local. So I would highly recommend a bit more nuance here. "The largest quantity of groundnut seeds distributed by Nakaseke Seedbank was for local varieties (line 245)." Please highlight the heterogeneity here between the three gene banks.

You could also mention that the contents of the seed banks in a way influence what is requested. In the conclusion you should outline your philosophy for how the seed banks should work. That is, should seed banks focus on 1) matching the demand of farmers? or 2) focus on conservation of local varieties or 3) some combination of these two? How should the seed banks consider expanding their collections? And provide more of a choice to farmers by for example running trials of creole, improved and landrace varieties of a given crop in the same plot to demonstrate their performance for farmers in the village?

The conclusion has been improved following your guidance as follows:

This study has moved community seed banking from the conceptual and theoretical sharing of experiences to showcasing the practicalities and realities of the concepts and theories. The results highlight the contribution of community seedbanks in securing farmers’ access to crop and seed diversity of local varieties /landraces, improved / modern and creole varieties. There was heterogeneity in the focus of the studied seedbanks on the different types of varieties whereby the bigger quantity of groundnut seeds distributed by Nakaseke seedbank was for local varieties yet for common bean; Rubaya and Kibuga seedbanks distributed bigger quantities of modern varieties while Nakaseke distributed the biggest quantity of creole varieties. This heterogeneity is good in that it enables the seedbanks to serve farmers under different socio-economic conditions. Since the contents in the seedbanks influence what is requested and accessed by farmers, seedbanks should be managed in such a way that they meet farmers’ demand in terms of quantities and varieties. However, seedbanks should put greater emphasis on conserving landraces to prevent their extinction as they are important sources of genetic material for breeding due to their wider adaptation to different conditions that enables them to withstand both abiotic and biotic stresses [35]. Community seed banks need to establish demonstration gardens and/or on-farm trials in their areas of operation; to assess the yield of landraces, creole and improved varieties on the same plot under the same management conditions; to enable farmers make the best variety choices for their conditions. And to widen the variety choice base, seedbanks should consider expanding their varieties collections through networking with public national genebanks, local and international seedbanks but also through their other social networks. There is need to improve the capacity of seedbanks in research and networking to enable more effective management of the crop genetic resources they handle.

The study has revealed that women are more enthusiastic and active in community seed banking, and they should be formally recognized, appreciated, and supported for their active role, knowledge, passion, and expertise in managing seedbanks. The results also show that the extent to which the core functions of genetic resources conservation and seed sovereignty were achieved  depended on the modes of operation of the seedbanks including:, actors management, degree of development and social-economic setting, as documented in [40].Further research is recommended to un-pack the inside of each of these factors and come up with the appropriate combinations of these factors that make the most conducive environment for effective community seed banking in particular contexts. In order have more comprehensive information on the role of the community seed banks, further research could include a social study through interviewing farmers to understand how they value seedbanks and how they would like to improve them to complement what is captured by the approach of our study.

Also, here are a few text edits: "Green Revolution" should be capitalized on line 43; "cultural" on line 68; "Access to" on line 112; "seed banks" on line 458. Please re-read carefully to proofread.

The text edits have been made and the manuscript proofread to eliminate any mistakes.

Thank you so much.

Sincerely,

Rose Nankya

Corresponding Author.
